# Job Satisfaction among Occupational Therapists Employed in Primary Care Services in Norway

**DOI:** 10.3390/ijerph20065062

**Published:** 2023-03-13

**Authors:** Tore Bonsaksen, Sissel Horghagen, Cathrine Arntzen, Astrid Gramstad, Linda Stigen

**Affiliations:** 1Department of Health and Nursing Sciences, Inland Norway University of Applied Sciences, 2418 Elverum, Norway; 2Department of Health, VID Specialized University, 4024 Stavanger, Norway; 3Department of Neuromedicine and Movement Science, Faculty of Medicine and Health Sciences, Norwegian University of Science and Technology, 7491 Trondheim, Norway; 4Department of Health and Care Sciences, UiT the Arctic University of Norway, 9037 Tromsø, Norway; 5Centre for Care Research, North, 9037 Tromsø, Norway; 6Department of Health Sciences Gjøvik, Norwegian University of Science and Technology, 2802 Gjøvik, Norway

**Keywords:** employee influence, health services, job satisfaction, national survey, occupational therapy, primary care, work engagement

## Abstract

The aging population will place healthcare services under considerable strain in the years ahead. Occupational therapists play a vital role in securing sustainable healthcare services and are increasingly employed by municipalities. To promote sustainable services, the job satisfaction among core professional groups needs monitoring. A comprehensive cross-sectional survey was distributed among municipality-employed occupational therapists in Norway during May–June 2022, to which 617 responded. Job satisfaction was assessed with the Job Satisfaction Scale (JSS), and factors associated with job satisfaction were assessed with linear regression analysis. The mean JSS score in the sample was 51.4. The regression model explained 14.4% of the variance in job satisfaction scores. Having more work experience as an occupational therapist (*β* = 0.16, *p* = 0.02) and having higher perceived influence on the work unit’s goals (*β* = 0.31, *p* < 0.001) were significantly related to higher job satisfaction. The study implies that job satisfaction in the occupational therapy profession increases with years of experience and also with the ability to engage with and influence the larger work environment. Thus, to promote job satisfaction, occupational therapists should seek to engage not only with their own work at hand but also with the larger goals and strategies of the organization they work for.

## 1. Introduction

In Norway, healthcare is changing from a hospital-based to a community-based model of care. Following the Coordination Act in 2012 [1], more tasks and responsibilities have been placed on municipalities while hospitals have shifted toward performing highly specialized services and procedures for people in need of high-intensity treatment. The changes in the organization of healthcare services have also brought about changes in the required composition of healthcare personnel in the municipalities, and the changes have led to healthcare personnel performing new tasks, for new user groups, in collaboration with new partners.

Community-working occupational therapists in Norway are involved in health promotion initiatives and rehabilitation of people with a variety of health conditions, supporting their functioning in activities of daily living and their participation in desired arenas in society [2]. Groups of services users include older people, people with mental health and substance use problems, and people with noncommunicable diseases and lifestyle-related health problems. Supporting people’s (re-)integration into society, including getting and holding on to a job, is increasingly emphasized in the profession [2]. Occupational therapists’ participation in multiprofessional collaboration is common, and in view of the relative shortage of health personnel as foreseen [3], it is likely to become an even more important aspect of occupational therapists’ work organization in the future. This development requires that occupational therapists understand the expectations that other personnel groups have regarding their role and that they are able to negotiate these expectations if needed [4]. The implementation of multiprofessional practice has been shown to create synergies that can assist in sustaining people’s health in local contexts [5].

For occupational therapists, 2020 marked the profession’s transition from an elective municipal health service to one required by law [6]. Over the recent years, and probably in part due to the change in legislation, there has been an increase in the number of occupational therapists working in the country’s municipalities [7]. The increase in numbers is also reflected in better coverage. The coverage of 3.7 occupational therapists per 10,000 inhabitants in 2015 increased to 4.8 per 10,000 inhabitants in 2021 [7].

While the evidence points to an increased number of occupational therapists in the municipalities, as well as to a better coverage of services, there seems to be less available knowledge concerning occupational therapists’ work environments and whether occupational therapists respond to these with higher or lower levels of job satisfaction. Job satisfaction, while loosely and ambiguously defined in previous work [8], broadly refers to employees’ positive and negative feelings regarding their job [9]. It can be applied as an all-encompassing concept, denoting the extent to which employees are generally satisfied with their job or refer to specific job dimensions. Thus, an employee’s job satisfaction may vary across dimensions and across time due to changes in the work environment and/or in the employee’s perception of the work environment [9]. In general, job satisfaction is likely to be higher for employees who perceive a high degree of autonomy [10], social and organizational support [11], and opportunities for skill development [12] and appropriate rewards, and lower for those who feel the job is stressful [13,14], unrewarding [15], and without appropriate support [11] and opportunities for autonomy [10] and development [12].

Worldwide, much variability in job satisfaction has been reported within and between different professional groups, and researchers have suggested that variations in culture, work conditions, and work expectations are the main drivers of these differences [9,16]. Similarly, studies of healthcare professionals have reported varying levels of job satisfaction. For example, higher levels have been found among physicians in Norway, compared to physicians in other countries [17,18,19]. Higher levels of job satisfaction have been found in most studies of physiotherapists [16,20,21], but not in all of them [22]. Studies of nurses have also yielded mixed results. A study from England found high levels of job satisfaction among nurses [23], whereas a study from Norway found overall job satisfaction to be at a moderate level [24]. A previous Norwegian study of occupational therapists found a very high level of job satisfaction [25]. However, a small sample size, geographically restricted recruitment, and the use of a nonvalidated outcome measure indicate that further research is warranted to corroborate the finding.

Further to investigating levels of job satisfaction within and between different groups of employees, several studies have used statistical models to estimate the strength of associations between measures of job satisfaction and a variety of predictor variables. For the purpose of this study, we suggest that such tentative predictor variables be grouped into person-related variables (such as sociodemographic characteristics), work structure variables (factors related to the organization of the work, such as whether one is located together with same-profession employees), and practice variables (factors more directly related to the content of the work, such as the number of clients seen daily).

Using the proposed model to frame previous research findings, it appears that all variable groups contribute to explain variations in job satisfaction. Among the person-related factors, higher job satisfaction has been found to be associated with higher age [19] and more years in higher education [25,26], while being ambiguously related to gender. Some studies have found higher satisfaction among women [22], while others have not detected any association [27,28]. In relation to work structure, job satisfaction has been found to be higher among therapists working in outpatient care, compared to those in hospital-based care [28], while results have been mixed or nonconclusive with regards to working in private or public practice [19,20,22]. Being integrated with, and appreciated by, co-workers from one’s own professional group and other immediate co-workers has been linked with higher levels of job satisfaction [23,29], whereas perceiving the profession to have poor status and profile have been described as reasons for job dissatisfaction [30]. Among practice-related factors, having a high workload and feeling overworked have been related to lower job satisfaction [19,26,31], while access to a good mentor [21,26] and perceiving the work to be both rewarding and cognitively challenging [32] have been linked with higher job satisfaction.

The shift toward a community-based approach to healthcare necessitates more research on the various requirements underpinning the organization and delivery of primary healthcare services. In a situation with increasing healthcare demands in the foreseeable future, sustainable healthcare services depend on a stable and well-functioning workforce of health professionals [3]. Dissatisfaction among workers may instigate poorer work performance as well as higher turnover rates. Thus, to ensure a sustainable workforce of occupational therapists in Norwegian primary care services, their levels of job satisfaction need to be monitored and upheld. Thus, the aim of the study was to examine levels of job satisfaction among occupational therapists employed in primary care services in Norway and to examine factors associated with job satisfaction in this group of employees.

## 2. Materials and Methods

The study is part of an ongoing investigation of occupational therapy practice, organization, and development in Norway between 2017 and 2022 [33,34,35,36,37,38,39]. The current study was designed as a cross-sectional quantitative study based on a national survey conducted among occupational therapists working in primary care services. The data were collected between 12 May and 10 June 2022.

### 2.1. Population and Sample

Of the 2122 eligible participants, 617 occupational therapists (29.1%) chose to participate in the survey. With regard to the gender distribution, 93.2% were female, 6.5% were male, and 0.3% identified as other. The age range spanned from 23 to 64 years, with a mean of 42.1 years (SD = 11.3 years). The mean age of the sample was similar to that of the municipality-working population listed as members of Ergoterapeutene (the Norwegian Association of Occupational Therapists, mean age: 43 years), whereas the sample had a somewhat lower proportion of males compared with the municipality-working members of Ergoterapeutene (12.7%).

### 2.2. Data Collection and Procedures

A survey was developed to explore different aspects of the practice and context of Norwegian community-based occupational therapists. The survey tool was initially developed in 2017; however, it was further developed prior to the 2022 data collection due to a desire to collect more detailed information about the content of the occupational therapists’ practices. The complete survey tool is available from the authors upon reasonable request. The specific survey items used in the current study are described in the Measures section (Section 2.3).

The questionnaire was piloted among seven occupational therapists representing two municipalities (serving 13,500 and 33,500 inhabitants) to ensure that all relevant response options were included and that the survey questions were unambiguous. Based on their feedback, the questionnaire was revised. The revisions included altering the numbering on some questions, changing the layout on two questions with multiple response options, adding some response options to one question, and changing one question with fixed response options into an open-ended question.

The member list of Ergoterapeutene was used to identify relevant participants. An invitation email containing a link to the electronic survey was sent to 2122 occupational therapists eligible for participation. Three reminders were sent in an attempt to reach nonresponders to the initial distribution. The reminders were sent one, two, and three weeks after the initial invitation was sent. The survey was closed after four weeks, at which point all data were transferred to the project group.

### 2.3. Measures

Job satisfaction was measured with the Job Satisfaction Scale (JSS) [40]. It is a 10-item scale allowing the respondent to provide a graded response to questions concerning the level of satisfaction with different aspects of the job. These aspects are concerned with responsibility; task variation; co-workers and other employees; physical work conditions; opportunity to use skills; the job overall, all things considered; freedom to choose one’s own work methods; recognition for a job well done; wages; and work hours. All items are rated 1–7, with 1 indicating the lowest level of satisfaction and 7 indicating the highest level. The scale has previously been translated into Norwegian and found to be a valid measure of job satisfaction [41]. Previous studies have demonstrated acceptable reliability estimates (Cronbach’s α between 0.79 and 0.85) [20,42]. The current study verified a one-factor structure by principal components analysis (Factor 1: ƛ = 5.05, 50.5% explained variance, all factor loadings > 0.40), and a high level of internal consistency between items (Cronbach’s α = 0.88). The JSS scale score range was 10–70.

Sociodemographic factors included age (years), gender (male/female/other), education level (B.Sc.-level training, further education, master’s degree, doctoral degree), and years of experience from working as an occupational therapist (continuous). All analyses using the gender variable excluded those identifying as ‘other’.

Work structure factors included number of occupational therapists (OTs) employed in the municipality (1 = 0–1 OTs, 2 = 2–3 OTs, 3 = 4–5 OTs, 4 = 6–7 OTs, 5 = 8–9 OTs, 6 = 10–11 OTs, 7 = 12–13 OTs, 8 = 14–15 OTs, 9 = more than 15 OTs), being located together with other occupational therapists (yes or no), job size (full job or not), team organization (occupational therapy service or combined occupational therapy/physiotherapy service versus multiprofessional/other forms of work organization), and line manager’s educational background (occupational therapist or not).

Practice factors included level of perceived influence on the work unit’s goals (1 = not at all, 2 = a little, 3 = somewhat, 4 = much, 5 = very much), number of clients seen on a daily basis (1 = 0–1 clients, 2 = 2–3 clients, 3 = 4–5 clients, 4 = 6 or more clients), proportion of work hours spent on tasks related to assistive aids (10%, 20%, 30%…100%), waitlist (‘yes’ versus ‘no’ or ‘unsure’), teamwork (1 = work mostly alone, 2 = work as much alone as with others, 3 = work mostly with others), and participation in research and development activities (‘yes’ versus ‘no’).

### 2.4. Data Analysis

All variables were explored with frequencies and percentages (for categorical variables) and mean and standard deviations (for continuous variables). The outcome measure (JSS) did not follow the normal distribution (Kolmogorov–Smirnov = 0.10, *p* < 0.001). Thus, bivariate associations with JSS were investigated with both parametric (Pearson’s *r*) and nonparametric (Spearman’s rho) tests, and the results were deemed similar. In light of the similar results obtained by the two methods, we proceeded with parametric analyses.

To avoid any suppressor effects, variables indicating a possible direct association (bivariate association *p* < 0.30) with the outcome were included in a hierarchical linear regression analysis. Following this procedure, the regression model was built in three blocks: (1) sociodemographic variables (age, gender, education, work experience), (2) work structure variables (number of occupational therapists in the municipality, located together with other occupational therapists, size of job position, job organization, line manager’s educational background), and (3) practice variables (perceived influence on the work unit’s goals, number of clients seen daily, time proportion spent working with assistive aids, waitlist, working alone or in a team, and participation in research and development). Based on Cohen [43], the strength of associations was interpreted from the standardized β values in final model: β about 0.10 is a small effect, β about 0.30 is a medium effect, and β about 0.50 is a large effect. Model fit was interpreted from the adjusted *r*^2^ for the final model. Associations demonstrating *p* < 0.05 were interpreted as statistically significant.

Deviation from the normal distribution on the outcome measure occurs frequently in large datasets and is normally not considered a threat to the validity of parametric tests [44]. In addition, skewness on the JSS was minor (−0.78, SE = 0.10) and well within the recommended margins [45]. In the regression analysis, all variance inflation factors (VIF) were below 1.5. The plot of standardized residuals was visually inspected, and they were deemed to approximate the normal distribution. However, the standardized residuals (ranging between −4.7 and 2.2) fell below the recommended range in the lower end [46], indicating that some caution should be exercised in the interpretation of the regression analysis results.

### 2.5. Research Ethics

Participants were informed that participation was voluntary and anonymous, and participating by entering the link provided in the invitation email was considered informed consent. Approval for the study was obtained from the Norwegian Centre for Research Data (project number 52827).

## 3. Results

### 3.1. The Sample

In the sample, there was a large majority of women (93.2%). The mean age was 42 years, and the mean length of experience from working as an occupational therapist was 14 years. While 46% had further education following their basic training as occupational therapists, only 11% had a postgraduate degree (master’s or doctoral degree). The mean JSS rating was 51.4 (SD = 10.0). The distribution of JSS ratings is shown in Figure 1, while the sample characteristics are shown in Table 1.

### 3.2. Unadjusted Associations with Job Satisfaction

In preparation for the regression analysis, the bivariate relationships between each of the independent variables and the JSS ratings were examined. Both parametric and nonparametric tests were used. Most associations showed a small effect size, while the association between perceived influence and job satisfaction was of medium size (r = 0.33, *p* < 0.001). Higher levels of job satisfaction were related to higher age, more years of experience, higher perceived influence, less time spent working with assistive aids, working in a team, and participation in research and development work. Unadjusted associations between independent variables and job satisfaction are shown in Table 2.

### 3.3. Adjusted Associations with Job Satisfaction

As revealed from the hierarchical linear regression analysis, most of the adjusted associations were not statistically significant. In the fully adjusted model, the analysis revealed statistically significant associations between higher job satisfaction and more years of experience as occupational therapist (β = 0.16, *p* = 0.02) and higher perceived influence over the work unit’s goals (β = 0.31, *p* < 0.001). The predictors in the final model accounted for 14.4% of the variation in JSS scores. The adjusted associations between the independent variables and job satisfaction are shown in Table 3.

## 4. Discussion

### 4.1. Main Summary of Findings

The mean level of job satisfaction (mean JSS rating) among the occupational therapists was 51.4. In the fully adjusted predictive model, higher levels of job satisfaction were associated with more years of work experience as occupational therapist and higher perceived influence over the work unit’s goals.

### 4.2. Level of Job Satisfaction Compared with Other Professional Groups

The mean JSS score in the sample was, judging by comparison with other professional groups, in the middle range. Compared to Norwegian nurses [24,42], the occupational therapists’ job satisfaction was higher, while it was lower compared to Norwegian physicians [18,19,42] and physiotherapists [16,20]. However, job satisfaction is demonstrably related not only to professional group but also to the specific context of work. For example, job satisfaction was considerably higher among Norwegian physicians compared to physicians from Germany [18], and there were differences between Norwegian physicians working in different specialties [19]. In addition, psychomotor physiotherapists in Norway who were employed in public practice had higher job satisfaction compared to their self-employed counterparts in private practice [20]. As seen in Figure 1, there were large variations in job satisfaction also among the occupational therapists working in primary care services, but this study was unable to identify any work structure variables or other specific contexts that might explain such variations. Future research may focus on job satisfaction among occupational therapists in different contexts.

### 4.3. Factors Associated with Job Satisfaction

More years of experience from occupational therapy practice was found to be related to higher job satisfaction. One might consider more years of experience to be a mere proxy for age, and higher age has been related to job satisfaction in some studies [19], while it was unrelated in others [18,20]. However, age was not significantly related to job satisfaction in the multivariate analysis, indicating that more years of relevant professional experience is distinctly different from personal maturity, which can be expected to increase with age. Possibly, more years practicing as an occupational therapist may yield more time to nurture a deeper interest in the strengths and possibilities embedded in the profession’s history, values, theories, and practices, which in turn may translate into higher satisfaction with the job. In support of this reasoning, a previous study found that a higher level of interest in the job was strongly associated with higher job satisfaction [25].

Perceived influence on the work unit’s goals was also significantly, and with moderate strength, related to higher levels of job satisfaction. The perception of having an influence extending beyond one’s own day-to-day practice may equate to a sense of belonging within the organization and a sense of being important and valuable for the organization. These are positive experiences that would be logically linked with perceptions of the job as meaningful and of being able to use one’s strengths and abilities. Such experiences may lead to higher levels of engagement [47], and previous studies of occupational therapists have pointed toward autonomy, belonging, and acknowledgement within a professional community as important facets of the job [33,38]. Feeling valued has been linked with lower burnout [48], whereas a lack of appreciation [49] and a heavy workload [50] have been described among precursors for considering to leave the profession.

### 4.4. Study Limitations

The cross-sectional nature of the data collected precludes any causal associations to be made. Thus, despite the tentative presentation in this paper, oppositely directed associations, or bicyclical self-strengthening associations, are viable. For example, more years of experience as an occupational therapist may foster higher levels of job satisfaction, whereas higher job satisfaction is also a probable cause for continuing a career as an occupational therapist. The response to mailed surveys is often relatively low [51,52]. The response rate in this study was also in the lower range, and somewhat lower than what was obtained in the previous survey conducted in 2017 [35]. However, the sample had a mean age very similar to that of the full group of members of Ergoterapeutene, whereas the sample’s proportion of male participants was somewhat lower. We have no further information about the degree of correspondence between the characteristics of sample and those of the population, which should be considered when generalizing from the sample to the population. While we used a validated and frequently employed measure of job satisfaction, several of the other measures used in the study were self-developed and without known measurement properties. The assumptions underpinning the regression analysis were not fully validated, as indicated by the spread of the standardized residuals. The regression model also explained a relatively low proportion of the outcome variance, indicating that future studies need to take other factors into account as they set out to explain job satisfaction among occupational therapists.

## 5. Conclusions

Job satisfaction among Norwegian occupational therapists in primary care services was found to be in the middle range: higher than found in previous studies of nurses, while lower than found in studies of physicians and physiotherapists. Higher job satisfaction was associated with more years of work experience and the perception of having more influence on the work unit’s goals. Thus, occupational therapy in primary care may be considered a type of practice where a deeper interest in the strengths and possibilities embedded in the profession and the ability to engage in strategic processes within the larger organization are important for the satisfaction derived from the job.

## Figures and Tables

**Figure 1 ijerph-20-05062-f001:**
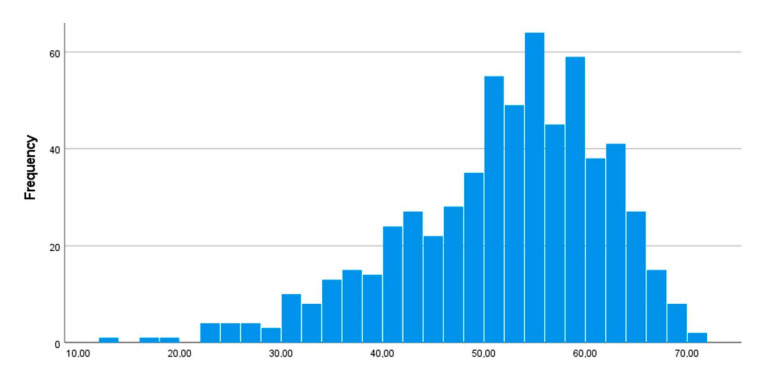
Job satisfaction (JSS ratings) in the sample.

**Table 1 ijerph-20-05062-t001:** Sample characteristics.

Variables	M (SD)	n (%)
*Sociodemographic variables*		
Age	42.1 (11.3)	
Female gender		575 (93.2)
Male gender		40 (6.5)
Years of experience as an OT ^1^	14.4 (9.8)	
B.Sc. level education		267 (43.3)
Further education		282 (45.7)
M.Sc. level education		66 (10.7)
Ph.D. level education		2 (0.3)
*Work structure variables*		
Number of OT positions in the municipality	5.2 (2.7)	
Located with other OTs		443 (71.8)
Not located with other OTs		174 (28.2)
Full job (100%)		523 (84.8)
Less than full job (<100%)		94 (15.2)
Job organized as OT service, or as OT and PT ^2^ service		326 (52.8)
Other type of (multiprofessional) organization		291 (47.2)
Line manager is an OT		132 (21.4)
Line manager is not an OT		485 (78.6)
*Practice variables*		
Perceived influence	2.7 (1.0)	
Number of clients seen daily	2.1 (0.7)	
Time proportion spent working with assistive aids	4.8 (3.1)	
Municipality has waitlist		448 (72.6)
Municipality does not have waitlist or ‘don’t know’		169 (27.4)
Works mostly alone		302 (48.9)
Works as much alone as in a team		187 (30.3)
Works mostly in a team		128 (20.7)
Participates in R&D ^3^ work		117 (19.0)
Does not participate in R&D work		500 (81.0)
Job satisfaction (JSS rating)	51.4 (10.0)	

Note. ^1^ OT = occupational therapist, ^2^ PT = physiotherapist, ^3^ R&D = research and development.

**Table 2 ijerph-20-05062-t002:** Unadjusted associations between independent variables and job satisfaction (JSS ratings).

Independent Variables	Job Satisfaction ^7^	
	*r*	rho
*Sociodemographic variables*		
Age	0.13 **	0.14 **
Gender ^1^	−0.07	−0.06
Years of experience as an OT ^1^	0.18 ***	0.17 ***
Education level ^2^	0.05	0.05
*Work structure variables*		
Number of OT positions in the municipality	−0.02	0.01
Located together with other OTs ^3^	0.05	0.05
Job size ^4^	0.08	0.07
Job organization ^5^	−0.06	−0.07
Line manager’s background ^6^	−0.03	−0.04
*Practice variables*		
Perceived influence ^7^	0.33 ***	0.36 ***
Number of clients seen daily ^7^	−0.03	0.01
Time proportion spent working with assistive aids ^7^	−0.12 **	−0.14 ***
Waitlist ^8^	−0.02	−0.05
Works mostly alone or in a team ^9^	0.14 ***	0.13 ***
Participation in R&D work ^10^	0.10 *	0.11 **

Note. ^1^ Female gender is higher value. ^2^ Higher education levels are higher values. ^3^ Located with other OTs is higher value. ^4^ Full job is higher value. ^5^ Job organized as OT service or combined OT and PT service is higher value. ^6^ Line manager having OT educational background is higher value. ^7^ Higher perceived influence, more clients seen daily, higher time proportion spent working with assistive aids, and higher job satisfaction are higher values. ^8^ Municipality with waitlist is higher value. ^9^ Works more often in a team is higher value. ^10^ Participation in R&D work is higher value. * *p* < 0.05, ** *p* < 0.01, *** *p* < 0.001.

**Table 3 ijerph-20-05062-t003:** Hierarchical linear regression analysis displaying adjusted associations between independent variables and job satisfaction (JSS ratings).

Independent Variables	Job Satisfaction ^6^	
	*β*	*p*
*Sociodemographic variables*		
Age	0.01	0.92
Gender ^1^	−0.06	0.14
Years of experience as OT	0.16	0.02
Education level ^2^	−0.02	0.57
**Adjusted *r*^2^**	**2.8%**	**<0.001**
*Work structure variables*		
Located with other OTs ^3^	0.07	0.08
Job size ^4^	0.06	0.13
Job organization ^5^	−0.02	0.60
**Adjusted *r*^2^**	**3.6%**	**<0.001**
*Practice variables*		
Perceived influence ^6^	0.31	<0.001
Time proportion spent working with assistive aids ^6^	−0.01	0.80
Works mostly alone or in a team ^8^	0.09	0.05
Participation in R&D work ^9^	0.03	0.49
**Adjusted *r*^2^**	**14.4%**	**<0.001**

Note. Adjusted *r*^2^ (bold) is the proportion of the variance in job satisfaction scores explained by the preceding variables in the model. ^1^ Female gender has higher value. ^2^ Higher education levels have higher values. ^3^ Located with other OTs has higher value. ^4^ Full job has higher value. ^5^ Job organized as OT service or combined OT and PT service has higher value. ^6^ Higher perceived influence, higher time proportion spent working with assistive aids, and higher job satisfaction have higher values. ^8^ Works more often in a team has higher value. ^9^ Participation in R&D work has higher value.

## Data Availability

The data presented in this study will by the research project’s completion become available on reasonable request from the corresponding author. The data are not publicly available due to ongoing dissemination and publication from the project.

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
