# Peer review of "Job Satisfaction among Occupational Therapists Employed in Primary Care Services in Norway"

_ijerph, 2023, doi:10.3390/ijerph20065062_

Round 1

Reviewer 1 Report

Tore Bonsaksen et al. submitted to IJERPH an article, focusing to the job satisfaction among OTs employed in primary healthcare services in Norway.

The topic is very interesting for healthcare workers, this manuscript is written with a scientific method, with an appropriate English language style and it is supported by adequate references, indicated in compliance with the Instructions for the Authors of IJERPH. There are numerous self-citations which, however, appear adequate.

Here are my suggestions:

LL 39-41: please, take the opportunity to briefly promote the topic of multiprofessionalism in healthcare services, considering the results coming from the following studies: DOI: 10.1186/s12875-021-01568-9   and   DOI: 10.3390/healthcare10101906

LL 42-43: please, detail what are the main tasks, roles and responsibilities of the OTs in your jurisdiction

- considering 2122 eligible participants, with 617 OTs responders, the minimum sample size, representative of the whole population, is largely exceeded: good!

- In line 111 it is written “The age range spanned from 23 to 64 years, with a mean of 43.8 years (SD=11.3 years)”, while analyzing table 1, it appears 42.1 +/- 11.3. Please align the data or clarify the discrepancy.

Thank you for your efforts in perfecting this important article!

Author Response

Authors: Thank you for the comments to the manuscript. We have modified the manuscript according to suggestions, with changes shown using track changes. We look forward to your feedback.

Reviewer 1 (R1): Tore Bonsaksen et al. submitted to IJERPH an article, focusing to the job satisfaction among OTs employed in primary healthcare services in Norway. The topic is very interesting for healthcare workers, this manuscript is written with a scientific method, with an appropriate English language style and it is supported by adequate references, indicated in compliance with the Instructions for the Authors of IJERPH. There are numerous self-citations which, however, appear adequate.

Authors: Thank you for the kind feedback.

R1: Here are my suggestions: LL 39-41: please, take the opportunity to briefly promote the topic of multiprofessionalism in healthcare services, considering the results coming from the following studies: DOI: 10.1186/s12875-021-01568-9   and   DOI: 10.3390/healthcare10101906

Authors: The Introduction section has been expanded to incorporate the topic of multi-professional work; see added materials line 50-57 in the revised manuscript. The suggested references have been included.

R1: LL 42-43: please, detail what are the main tasks, roles and responsibilities of the OTs in your jurisdiction

Authors: We have added a section in response to this comment, see line 43-49 in the revised manuscript.

R1: considering 2122 eligible participants, with 617 OTs responders, the minimum sample size, representative of the whole population, is largely exceeded: good!

Authors: Thank you for the kind feedback.

R1: In line 111 it is written “The age range spanned from 23 to 64 years, with a mean of 43.8 years (SD=11.3 years)”, while analyzing table 1, it appears 42.1 +/- 11.3. Please align the data or clarify the discrepancy.

Authors: Thank you for pointing out the error; corrected on line 128 in the revised manuscript.

R1: Thank you for your efforts in perfecting this important article!

Authors: Thank you for the kind feedback.

Reviewer 2 Report

Job satisfaction is an important topic to be researched. Given the changes in the Norwegian health system from a hospital-based to a more community-based health care system, this article focuses on an important are. The Norwegian change is potentially also pioneering for other countries around the world.

The article is a lean on a conceptual basis. What exactly job satisfaction means, could be defined a little more clearly (see e.g. Aziri B. JOB SATISFACTION: A LITERATURE REVIEW MANAGEMENT RESEARCH AND PRACTICE VOL. 3 ISSUE 4 (2011) PP: 77-86).  A brief discussion of what job conditions or structures, colleagues' behavior etc. are counterproductive to job satisfaction would be helpful for the reader.

I would also encourage the authors to maybe shift some of the finding presented in the introduction (second last para) to the conclusion which is otherwise very short.

Great work otherwise.

Author Response

Authors: Thank you for the comments to the manuscript. We have modified the manuscript according to suggestions, with changes shown using track changes. We look forward to your feedback.

Reviewer 2 (R2): Job satisfaction is an important topic to be researched. Given the changes in the Norwegian health system from a hospital-based to a more community-based health care system, this article focuses on an important are. The Norwegian change is potentially also pioneering for other countries around the world.

Authors: Thank you for the kind feedback.

R2: The article is a lean on a conceptual basis. What exactly job satisfaction means, could be defined a little more clearly (see e.g. Aziri B. JOB SATISFACTION: A LITERATURE REVIEW MANAGEMENT RESEARCH AND PRACTICE VOL. 3 ISSUE 4 (2011) PP: 77-86).  

Authors: We have elaborated more on the job satisfaction concept, including the suggested reference, see line 68-74 in the revised manuscript.

R2: A brief discussion of what job conditions or structures, colleagues' behavior etc. are counterproductive to job satisfaction would be helpful for the reader.

Authors: More details, including a range of new references, have been provided to highlight factors both promoting and hindering job satisfaction, see line 74-79 in the revised manuscript.

R2: I would also encourage the authors to maybe shift some of the finding presented in the introduction (second last para) to the conclusion which is otherwise very short.

Authors: The findings reported in the relevant introduction section are those of previous research, hence they cannot be presented in the concluding section of the current study.

R2: Great work otherwise.

Authors: Thank you for the kind feedback.